# Good Things in Small Packages? Evaluating an Economy of Scale Approach to Behavioral Health Promotion in Rural America

**John P. Bartkowski [1],\* , Xiaohe Xu [2,3] , Jerri S. Avery [4], Debbie Ferguson [5] and Frankie J. Johnson [6]**

1   Department of Sociology, The University of Texas at San Antonio, San Antonio, TX 78249, USA
2   School of Public Administration, Sichuan University, Chengdu 610065, Sichuan, China; xiaohe.xu@utsa.edu
3   Department of Sociology, The University of Texas at San Antonio, San Antonio, TX 78249, USA
4   Burton & Associates, Madison, MS 39110, USA; jerriavery@gmail.com
5   School of Advanced Studies, University of Phoenix, Phoenix, AZ 85040, USA; debbieferguson72@yahoo.com
6   Central Mississippi Residential Center, Newton, MS 39345, USA; fjay3@juno.com
\*   Correspondence: john.bartkowski@utsa.edu; Tel.: +1-210-508-2530

**Abstract:** Rural American youth exhibit pronounced health disparities. This study enlists insights from an economy of scale paradigm to determine the relative effects of serving smaller versus larger client groups in an assembly-style school-based behavioral health promotion program. Evaluation results are reported from a three-year intervention delivered to eighth-grade and tenth-grade rural Mississippi students from 2012 to 2015. The program, I Got U: Healthy Life Choices for Teens, coupled a day-long intensive immersion in youth risk prevention and mental health promotion with school-based information dissemination. Results reveal robust effectiveness in program years 1 and 2, during which caps of 175 attendees per event were imposed. Salutary results were no longer evident during year 3, when larger venues were used to serve over three times the number of students per event. This program teaches valuable lessons about the potential for diminishing returns yielded by an economy of scale approach to implementation.

**Keywords:** behavioral health; rural; South; Mississippi; youth; economy of scale; evaluation

## 1. Introduction

Youth risk prevention programs have become legion in the United States. With greater reliance on evidence-based programs, transformative policy changes such as the Government Performance and Results Act (GPRA), and frequently mandated National Outcome Measures (NOMs), there is now an increased emphasis on the rigorous evaluation of programs that had previously escaped scientific scrutiny. At the same time, there is a heightened awareness about health disparities, that is, the distinctive mortality and morbidity risks faced by disadvantaged populations [1–3]. In an effort to diminish or eradicate these disparities, intervention efforts are now often strategically targeted at high-risk populations.

Given the magnitude of health disparities, there is also increasing pressure to scale up interventions so that they can serve the maximum number of clients with a limited pool of resources [4]. This emphasis on expanded capacity and efficiency gains follows closely from an economy of scale approach to organizational functioning and service delivery. An economy of scale model is designed to deliver cost savings that can be obtained through a number of means, most often through the growth (increased volume) of an enterprise. In a conventional economic sense, fixed production costs (e.g., labor, overhead) distributed over greater output (e.g., units produced) create an economy of

scale. Efforts have been undertaken to integrate economies of scale into health care delivery, with mixed results. For example, economies of scale can free up resources for significant gains in public health [5]. However, where medical service provision is concerned, economies of scale work best if agility (flexibility, rapid response, and synchronization) is coupled with the more common focus on leanness (operational convergence, service non-duplication) [6]. While economies of scale have been used in medical service delivery, research on behavioral health service economies of scale remains scant despite calls for action grounded in the principles of this approach [4,7]. Among its other aims, the present study is designed to contribute to this nascent literature.

Populations at risk of health disparities and other adversities are easily found in the rural South. Poverty rates in the rural South are among the highest in the nation, with greater percentages of impoverished children and adults living in such locales than anywhere else in the United States [8,9]. These problems are coupled with economic challenges (e.g., unemployment), educational deficits (e.g., inadequate funding of schools), food insecurity and food deserts, health care coverage gaps, and other adverse conditions that are exacerbated in rural southern communities [10,11]. Consequently, it is no surprise that significantly poorer health outcomes are observed among rural southern residents [12,13], with especially egregious health disparities evident in rural Mississippi [14–21].

Such adverse outcomes in Mississippi are not restricted to physical health conditions, but also include behavioral health risks such as high rates of unmet mental health needs and low rates of mental health service utilization [22]. Mississippi therefore follows other rural areas in exhibiting pronounced mental health deficits and behavioral health disparities, all of which are especially evident among young people within the state [23]. Moreover, the already pronounced behavioral health disparities in Mississippi have been magnified by natural and technological disasters such as Hurricane Katrina and the Deepwater Horizon oil spill, with such events exhibiting long-term negative impacts on mental health, particularly among the state's young and most disadvantaged residents [24–26]. But amidst these troubling circumstances resides an enticing opportunity. Rural communities within Mississippi are an excellent "natural laboratory" inasmuch as they can stringently test the effectiveness of an intervention designed to avert social risk.

Given these factors, a program was developed to foster improved behavioral health among youth living in rural communities in central Mississippi. The program, I Got U: Healthy Life Choices for Teens (hereafter, IGU), coupled a day-long intensive immersion in youth risk prevention and mental health promotion with school-based information dissemination related to self-esteem, alcohol and drugs, self-injury, suicide, mental health, and bullying. The Health Resources and Services Administration (HRSA) funded IGU under a three-year grant from 2012 to 2015. The IGU program was developed to serve rural eighth-graders and tenth-graders living in central Mississippi during its three-year funding cycle. Over 7000 youth were served during IGU's three-year funded implementation period, with a total of 5807 participating in the program evaluation. Risk assessment data on the pretest survey, available from authors upon request and discussed more fully in the Methods section that follows, indicated that the program's objectives aligned well with student needs. In every funding year, speakers were enlisted to deliver modules during each day-long IGU event. Speakers and programmatic content remained consistent throughout the three years of implementation. The IGU curriculum can be purchased by contacting this article's last author (F.J.J.). The evaluation instruments are included with the curriculum.

During the first two program years, IGU intentionally kept the number of student attendees capped at 175 per event. During year 3, increased demand for the intervention and the plan to apply for renewed funding led the project team to adopt an economy of scale approach. However, offering more events was not possible because the key supply conduit for the program—namely, the slate of featured speakers—could not increase their time commitment to the intervention. Therefore, in year 3, the program used larger venues to serve three times the typical number of students per event, with the largest event serving 800 students. While programmatic content and the slate of speakers did not change during implementation, the more than 4000 students served during year 3 was four times the

number of year 1 participants despite no appreciable change in the number of events held. The project team also believed that this change was warranted given increased speaker proficiency commonly exhibited in the late stages of an intervention. Given the single implementation change of increased participant numbers per event during the program's final year, IGU provides a good test case of the effectiveness of an economy of scale model (year 3) when compared with a more circumscribed approach to service delivery (years 1 and 2).

The section that immediately follows reports on the methods used to conduct the IGU evaluation, after which the results of the evaluation are conveyed. In brief, evaluation results were robust during the first and second project years. However, program expansion during year 3—that is, the economy of scale approach—significantly undermined IGU's effectiveness. During year 3, very few desirable outcomes were observed. Overall, the implementation and evaluation of IGU illustrate the need for great care when transitioning to scaled-up programming.

## 2. Methods

The IGU program was evaluated using pretest, post-test, and follow-up instruments featuring identical items to generate comparisons across time. Because IGU is a single-day immersive intervention, pretests and post-tests were administered on the same day, respectively, before and after the event. These pretest and post-test surveys were coupled with a follow-up survey that was administered four to six months after the intervention date. Eligibility for participation in the evaluation was defined by presence at the intervention and willingness to complete the survey. All participants were told of the voluntary nature of their participation in the evaluation and were assured that there would be no loss of benefits associated with a refusal to complete evaluation surveys.

The evaluation instruments featured measures of self-esteem, as well as attitudes toward bullying, drug use, suicide ideation, mental health stigma, and related risk factors, all of which were carefully piloted and validated prior to their use in the evaluation. Given the single-day nature of the intervention, behavioral measures could not be used on the pretest or post-test surveys. (Behavioral change would not be evident through a same-day administration of pretests and post-tests). To encourage student completion of the evaluation instruments, single-item indicators were used. In this way, each instrument could be contained on one side of a single 8.5-inch by 11-inch page. Evaluation instruments used during IGU were designed by a Ph.D.-level evaluation team with more than twenty years of combined experience in the field.

We generated and pretested a series of measures with strong face validity. The instruments were therefore composed of our own proprietary measures. Using scales of such constructs as self-esteem or mental illness stigma would have created an undue data collection burden on youth participants. Our effort to reduce the evaluation burden on youth participants while maximizing response rates led us to develop a single-page instrument. As events grew larger, this effort at response rate maximization was not as efficacious (comparative survey attrition rates yielding incomplete data increased in project years 2 and 3, and are discussed further in the Results section). There is value in using proprietary measures for program evaluation due to the more direct testing of key intervention elements. The instruments are available from the first author by request. There may be future opportunities for reliability testing as this intervention serves more students, Mississippians and non-Mississippians alike, if the program is adopted by other behavioral health agencies.

Instrument measures were generated based on the series of modules that, taken together, form the intervention. These modules were delivered as discrete elements of the intervention. Given the delivery of programmatic content in discrete modular form, our measures were created and analyzed as independent constructs designed to test the effectiveness of distinct parts of the intervention. Therefore, there were compelling epistemological and methodological rationales for analyzing the measures separately rather than grouping them through such techniques as factor analysis. In short, our analytical approach was designed to mirror program delivery.

Data on social risk factors evident among participants who completed the evaluation pretest instrument indicated several different areas of vulnerability, including academic performance problems, peer pressure, family problems, stress and anxiety, low self-esteem, and body image problems. Over the course of the three-year program, roughly equal numbers of male students (51%) and female students (49%) completed the pretest survey. African American students were well represented (42% of survey completion participants), which was roughly equivalent to the proportion of white students who participated (44%). Somewhat more eighth-graders were served (57%), when compared with tenth-graders (43%). Modal ages were 14 years old (27% of participants) and 16 years old (23% of participants).

Survey administration and data management protocols were affected by the relatively large size of the participant groups. For a behavioral health promotion class of 25–30 students, linking pretest, post-test, and follow-up surveys with a unique code assigned to each student is feasible. Yet, such is not the case with an intervention that serves several hundred people. Thus, even during years 1 and 2 when these assembly-style events were intentionally kept smaller, a 175-attendee cap per event prevented the use of respondent-matched surveys. Consequently, the evaluation team used unmatched surveys wherein it was not possible to link a corresponding pretest, post-test, and follow-up survey to the particular participant who completed them. (This limitation is discussed more fully in the study's conclusion.)

An additional limitation is presented by the size of survey batches, which varied somewhat across instrument administrations. As is common in the evaluation of rather large interventions, pretest surveys collected during the three-year intervention ($N = 5807$) were more abundant than the number of post-test surveys ($N = 3959$) and follow-up surveys ($N = 3051$) that were collected. Thus, some attrition in survey completion was exhibited across batches of pretest, post-test, and follow-up surveys. When compared with the completed number of pretests, there was 68% retention in the evaluation at post-test and 53% retention at follow-up.

Independent samples *t*-tests were used to conduct the analyses of raw data as collected through the evaluation surveys using four-point Likert-scale responses provided by students. Tables featured in the Results section show collapsed levels of student agreement with survey items for parsimonious presentation and ease of interpretation, but tests of statistical significance were conducted using the raw responses to ordinal measures. We used one-tailed tests for these data analyses. In statistical hypothesis testing, the null hypothesis is founded on the assumption that there is no reason to anticipate a significant association between two variables. So, when there is not a compelling alternative to the null hypothesis, a two-tailed test should be used because such a test will detect either a negative or positive association between the variables in question. Our situation presents a different scenario. Given the delivery of this intervention, which was piloted prior to our evaluation of it, we expected the effect to be in one particular direction. We reasonably expected the manifestation of improved outcomes when post-implementation responses were compared to their pre-implementation counterparts. One-tailed tests are more commonly used in intervention-based evaluations. The advantage of adopting this one-tailed test strategy is an improvement in power to reject the null hypothesis given the change-focused intent of intervention-based research, especially when programs have been piloted as had this intervention. The legitimate use of a one-tailed test is bolstered by the content of our intervention, particularly with regard to post-tests. We anticipated that students who participated in this day-long immersive program could retain the information very well on that same day, thus demonstrating improved results in the post-test.

Controls are not applied for the results presented here, but no evidence of confounding factors was observed based on ancillary analyses of the data. In all evaluation survey tables presented below, the figure featured in the "% Change" column is calculated by subtracting the percent of participants that agree at time 2 (T2, post-test or follow-up) from the percent that agree at time 1 (T1, pretest) divided by the percent that agree at time 1, based on the following equation: $[(T2 - T1)/T1] \times 100$.

This figure therefore represents a proportional percent change (relative to baseline) rather than a raw percent change.

It is important to note that the relative levels of overall agreement with each survey item displayed in the Results section tables are shown as a single collapsed "agree" category solely for ease of presentation and interpretation. Raw ordinal data based on actual student responses ranging from strongly agree to strongly disagree were used to conduct tests of statistical significance. The agreement categories were collapsed principally to offer a space-economizing effort in the tables. No response categories were recoded or collapsed for the purposes of data analysis because doing so would limit variability in these critical measures.

This evaluation was conducted with three notable limitations. First, a formal control group was cost-prohibitive for this intervention. The incentives needed to reward and retain control group members are costly, and the emphasis in this intervention was on using funds to deliver the program. Therefore, a conventional control group could not be used in this evaluation. Hence, while every effort was undertaken to ensure that rigorous methods were utilized in the context of this study, statistical comparisons could only be generated for what would typically be considered the treatment group (program attendees) over time.

Second, given the size of the groups to which the program was delivered and our prioritization of confidentiality with respect to student data, no identifiers were placed on any evaluation surveys. With no identifiers (even numeric codes) on any surveys, unmatched surveys at pretest, post-test, and follow-up were used to conduct the analyses featured here. We recognize the limits of this approach (matched surveys create greater comparability across survey waves), but the use of unmatched surveys was warranted for logistical and data security reasons.

Finally, we were unable to test for a possible cohort effect through which some students in a three-year program could have participated in year 1 (as eighth-graders) and year 3 (as tenth-graders). In our effort to minimize the data collection burden placed on youth participants with a single-page instrument, we did not track prior possible exposure to IGU. Of course, the prospect for a cohort effect is likely diluted in program year 3 due to program expansion, during which many more students were served. Thus, any possible cohort effect should be minimal, but it is a possible confounding influence worth noting.

## 3. Results

What, then, do the results of the IGU evaluation indicate? This section presents statistical results generated from the IGU evaluation in the following order. First, pretest/post-test comparisons of IGU are featured, respectively, for year 1 (Table 1), year 2 (Table 2), and year 3 (Table 3). Recall that year 3 represents the attempt to adopt an economy of scale approach to program delivery wherein approximately three times more participants attended each event while other implementation factors remained constant. Second, pretest/follow-up comparisons are featured for each program year (Tables 4–6). As a reminder, the "agree" column featured in these tables is provided to facilitate presentation convenience and accessible interpretation, particularly among non-scientists who sometimes read evaluation studies. No responses were recoded prior to statistical analyses. Raw data as ascertained directly from the surveys were used to conduct all analyses.

Turning first to pretest/post-test comparisons evident during year 1 of the program, Table 1 reveals ten statistically significant results from among the fifteen attitudinal survey items. All of the statistically significant results operate in the anticipated direction of beneficial (salutary) change from pretest to post-test. The most highly significant ($p < 0.001$) salutary changes were observed for attitudinal measures gauging self-esteem (item 1), willingness to encourage a friend engaged in self-harm to seek help (item 3), understanding the dynamics of healthy romantic relationships (item 8), recognizing the risk of harm associated with synthetic drugs (item 12), and understanding the real causes of depression (item 13). More modestly significant results were observed for participants' willingness to intervene against bullying (item 5, $p < 0.01$), disapproval attitudes toward peer marijuana use (item

7, *p* < 0.05), willingness to befriend a person with a mental illness (item 10, *p* < 0.01), recognizing the availability of an adult confidant (item 14, *p* < 0.05), and awareness of online bullying (item 15, *p* < 0.01). Taken together, significant positive changes on two-thirds of all indicators underscore the short-term success (at post-test) achieved during the first project year. It bears reiterating that the participant groups were intentionally kept small during project year 1. The tight attendee caps and small venues were designed to elicit a great deal of rapport and interaction between the speakers and students.

For project year 2, strong results were again evident from pretest to post-test (Table 2). Statistically significant salutary results were observed for seven of the fifteen survey outcomes, which is a noteworthy achievement. Highly significant changes (*p* < 0.001) were evident for understanding the dynamics of healthy romantic relationships (item 8) and awareness of online bullying (item 15). More modest but nevertheless significant results were apparent for willingness to intervene against bullying (item 5, *p* < 0.05), willingness to befriend a person with a mental illness (item 10, *p* < 0.01), awareness of the real causes of depression (item 13, *p* < 0.05), and having an adult confidant (item 14, *p* < 0.01). So, while statistically significant changes were more modest for year 2, this achievement again shows favorably on the intervention and its circumscribed approach (attendee caps, small venues).

**Table 1.** Pretest vs. post-test changes in I Got U (IGU) participant attitudes, project year 1.

| Items | Pretest | | Post-test | | Pre vs. Post | |
|---|---|---|---|---|---|---|
| | *n* | % Agree | *n* | % Agree | % Change | Sig |
| 1. I feel that I have a number of good qualities. | 1147 | 95.29 | 1013 | 96.84 | 1.63 | *** |
| 2. People my age who drink alcohol are hurting themselves. | 1166 | 86.62 | 1026 | 87.82 | 1.38 | |
| 3. I would encourage a friend who is cutting to get help. | 1157 | 90.67 | 1024 | 93.85 | 3.51 | *** |
| 4. Suicide sometimes seems like a good way to solve my problems. | 1155 | 14.73 | 1022 | 13.80 | −6.36 | |
| 5. If I saw someone getting bullied, I would speak up to stop the bullying. | 1156 | 89.45 | 1020 | 92.75 | 3.69 | ** |
| 6. I am hopeful about the future. | 1160 | 92.76 | 1021 | 92.46 | −0.32 | |
| 7. It's OK with me if kids my age are smoking pot (marijuana). | 1161 | 14.64 | 1019 | 12.37 | −15.55 | * |
| 8. True commitment to a boy-/girlfriend means having no other friends. | 1153 | 9.80 | 1019 | 8.83 | −9.88 | *** |
| 9. Smoking cigarettes and chewing tobacco are very harmful. | 1159 | 79.21 | 1021 | 80.80 | 2.02 | |
| 10. I would not be friends with someone who has a mental illness. | 1158 | 21.07 | 1021 | 16.75 | −20.51 | ** |
| 11. Doing well in school is important to me. | 1161 | 91.39 | 1023 | 92.67 | 1.40 | |
| 12. Spice (K2), bath salts, and drugs like these are safe to use. | 1141 | 12.81 | 1019 | 8.73 | −31.80 | *** |
| 13. Depression is not real; it's just an excuse for personal failings. | 1153 | 22.29 | 1012 | 16.30 | −26.85 | *** |
| 14. I could talk to an adult if I had a serious problem in my life. | 1159 | 80.24 | 1020 | 81.96 | 2.14 | * |
| 15. It's possible to be bullied online (like on Facebook or Twitter). | 1159 | 87.75 | 1023 | 89.83 | 2.38 | ** |

Statistical significance levels: * *p* < 0.05; ** *p* < 0.01; *** *p* < 0.001. Source: I Got U evaluation surveys, 15 attitudinal outcome measures.

**Table 2.** Pretest vs. post-test changes in IGU participant attitudes, project year 2.

| Items | Pretest | | Post-test | | Pre vs. Post | |
|---|---|---|---|---|---|---|
| | *n* | % Agree | *n* | % Agree | % Change | Sig |
| 1. I feel that I have a number of good qualities. | 1579 | 95.12 | 1110 | 94.86 | −0.27 | ** |
| 2. People my age who drink alcohol are hurting themselves. | 1605 | 85.55 | 1116 | 87.37 | 2.13 | |

**Table 2.** *Cont.*

| Items | Pretest | | Post-test | | Pre vs. Post | |
|---|---|---|---|---|---|---|
| | *n* | % Agree | *n* | % Agree | % Change | Sig |
| 3. I would encourage a friend who is cutting to get help. | 1599 | 92.56 | 1110 | 91.35 | −1.30 | |
| 4. Suicide sometimes seems like a good way to solve my problems. | 1588 | 12.85 | 1109 | 11.90 | −7.35 | |
| 5. If I saw someone getting bullied, I would speak up to stop the bullying. | 1585 | 90.09 | 1109 | 90.80 | 0.79 | * |
| 6. I am hopeful about the future. | 1581 | 93.86 | 1109 | 94.05 | 0.20 | |
| 7. It's OK with me if kids my age are smoking pot (marijuana). | 1583 | 17.75 | 1115 | 18.30 | 3.07 | |
| 8. True commitment to a boy-/girlfriend means having no other friends. | 1585 | 11.86 | 1106 | 10.22 | −13.86 | *** |
| 9. Smoking cigarettes and chewing tobacco are very harmful. | 1591 | 79.82 | 1105 | 81.36 | 1.92 | |
| 10. I would not be friends with someone who has a mental illness. | 1572 | 17.62 | 1105 | 15.48 | −12.18 | ** |
| 11. Doing well in school is important to me. | 1584 | 96.21 | 1104 | 95.11 | −1.15 | |
| 12. Spice (K2), bath salts, and drugs like these are safe to use. | 1571 | 8.59 | 1104 | 9.51 | 10.68 | |
| 13. Depression is not real; it's just an excuse for personal failings. | 1564 | 20.40 | 1093 | 19.58 | −4.01 | * |
| 14. I could talk to an adult if I had a serious problem in my life. | 1579 | 79.35 | 1098 | 83.33 | 5.01 | ** |
| 15. It's possible to be bullied online (like on Facebook or Twitter). | 1582 | 87.55 | 1106 | 90.69 | 3.59 | *** |

Statistical significance levels: * $p < 0.05$; ** $p < 0.01$; *** $p < 0.001$. Source: I Got U evaluation surveys, 15 attitudinal outcome measures.

**Table 3.** Pretest vs. post-test changes in IGU participant attitudes, project year 3.

| Items | Pretest | | Post-test | | Pre vs. Post | |
|---|---|---|---|---|---|---|
| | *n* | % Agree | *n* | % Agree | % Change | Sig |
| 1. I feel that I have a number of good qualities. | 2670 | 95.36 | 1629 | 95.33 | −0.02 | |
| 2. People my age who drink alcohol are hurting themselves. | 2686 | 85.26 | 1638 | 86.87 | 1.90 | |
| 3. I would encourage a friend who is cutting to get help. | 2674 | 92.37 | 1634 | 92.72 | 0.37 | |
| 4. Suicide sometimes seems like a good way to solve my problems. | 2674 | 11.93 | 1629 | 13.87 | 16.29 | |
| 5. If I saw someone getting bullied, I would speak up to stop the bullying. | 2673 | 91.77 | 1632 | 93.14 | 1.49 | |
| 6. I am hopeful about the future. | 2672 | 93.56 | 1628 | 92.38 | −1.26 | |
| 7. It's OK with me if kids my age are smoking pot (marijuana). | 2677 | 16.32 | 1632 | 16.30 | −0.15 | |
| 8. True commitment to a boy-/girlfriend means having no other friends. | 2668 | 10.19 | 1624 | 12.13 | 18.99 | |
| 9. Smoking cigarettes and chewing tobacco are very harmful. | 2672 | 81.10 | 1628 | 82.37 | 1.57 | |
| 10. I would not be friends with someone who has a mental illness. | 2665 | 18.12 | 1620 | 21.85 | 20.57 | ** |
| 11. Doing well in school is important to me. | 2678 | 93.61 | 1626 | 91.14 | −2.64 | ** |
| 12. Spice (K2), bath salts, and drugs like these are safe to use. | 2651 | 9.73 | 1617 | 11.50 | 18.19 | |
| 13. Depression is not real; it's just an excuse for personal failings. | 2649 | 21.93 | 1618 | 22.06 | 0.60 | |

**Table 3.** *Cont.*

| Items | Pretest | | Post-test | | Pre vs. Post | |
|---|---|---|---|---|---|---|
| | *n* | % Agree | *n* | % Agree | % Change | Sig |
| 14. I could talk to an adult if I had a serious problem in my life. | 2672 | 80.58 | 1624 | 82.70 | 2.63 | |
| 15. It's possible to be bullied online (like on Facebook or Twitter). | 2670 | 89.06 | 1626 | 91.02 | 2.20 | * |

Statistical significance levels: * $p < 0.05$; ** $p < 0.01$; *** $p < 0.001$. Source: I Got U evaluation surveys, 15 attitudinal outcome measures.

Project year 3 marked a significant shift in the delivery of the intervention, whereby attendee caps were lifted to serve over three times the number of IGU participants per event with no appreciable increase in the number of events due to speaker availability limitations. The programmatic content and slate of speakers remained the same, although the large student audiences during year 3 required a venue change in which presenters were on stage, thereby limiting interaction between them and the larger groups of student participants in the auditoriums to which the events were moved. What, then, were the pretest/post-test participant results associated with this final year during the effort to scale up the intervention? In a word, the results of the economy of scale approach were lackluster.

As revealed in Table 3, only one single survey item was statistically significant in a salutary direction, namely, online bullying awareness (item 15, $p < 0.05$). In fact, the only other statistically significant results observed for year 3 (items 10 and 11) indicate adverse changes when student responses before and after the intervention are compared. After the intervention, students were more inclined to agree that they would not befriend a person with a mental illness than they were beforehand. They were also less inclined to agree that doing well in school is important to them. It is, of course, possible that these adverse changes are related to the use of unmatched pretest and post-test survey items, given the possibility of selectivity bias at post-test. But what is more instructive is the general lack of significant salutary results in Table 3. In summary, a program that had been very successful during year 1 and still quite successful in year 2 exhibited a significant diminishment in its effectiveness during year 3, at least where evaluation outcome measures were concerned. At post-test, the economy of scale approach adopted in year 3 must be deemed unsuccessful.

**Table 4.** Pretest vs. follow-up changes in IGU participant attitudes, project year 1.

| Items | Pretest | | Follow-Up | | Pre vs. Follow-Up | |
|---|---|---|---|---|---|---|
| | *n* | % Agree | *n* | % Agree | % Change | Sig |
| 1. I feel that I have a number of good qualities. | 1147 | 95.29 | 729 | 95.61 | 0.33 | *** |
| 2. People my age who drink alcohol are hurting themselves. | 1166 | 86.62 | 733 | 81.72 | −5.66 | ** |
| 3. I would encourage a friend who is cutting to get help. | 1157 | 90.67 | 734 | 90.19 | −0.52 | |
| 4. Suicide sometimes seems like a good way to solve my problems. | 1155 | 14.73 | 727 | 13.62 | −7.56 | |
| 5. If I saw someone getting bullied, I would speak up to stop the bullying. | 1156 | 89.45 | 729 | 91.08 | 1.83 | |
| 6. I am hopeful about the future. | 1160 | 92.76 | 728 | 92.58 | −0.19 | |
| 7. It's OK with me if kids my age are smoking pot (marijuana). | 1161 | 14.64 | 729 | 23.05 | 57.38 | *** |
| 8. True commitment to a boy-/girlfriend means having no other friends. | 1153 | 9.80 | 723 | 11.48 | 17.14 | |
| 9. Smoking cigarettes and chewing tobacco are very harmful. | 1159 | 79.21 | 728 | 75.28 | −4.96 | * |
| 10. I would not be friends with someone who has a mental illness. | 1158 | 21.07 | 730 | 18.63 | −11.58 | |
| 11. Doing well in school is important to me. | 1161 | 91.39 | 729 | 90.67 | −0.78 | |

**Table 4.** *Cont.*

| Items | Pretest | | Follow-Up | | Pre vs. Follow-Up | |
|---|---|---|---|---|---|---|
| | *n* | % Agree | *n* | % Agree | % Change | Sig |
| 12. Spice (K2), bath salts, and drugs like these are safe to use. | 1141 | 12.81 | 731 | 9.44 | −26.30 | *** |
| 13. Depression is not real; it's just an excuse for personal failings. | 1153 | 22.29 | 728 | 21.43 | −3.86 | |
| 14. I could talk to an adult if I had a serious problem in my life. | 1159 | 80.24 | 730 | 77.40 | −3.55 | |
| 15. It's possible to be bullied online (like on Facebook or Twitter). | 1159 | 87.75 | 732 | 88.12 | 0.42 | |

Statistical significance levels: * $p < 0.05$; ** $p < 0.01$; *** $p < 0.001$. Source: I Got U evaluation surveys, 15 attitudinal outcome measures.

**Table 5.** Pretest vs. follow-up changes in IGU participant attitudes, project year 2.

| Items | Pretest | | Follow-Up | | Pre vs. Follow-Up | |
|---|---|---|---|---|---|---|
| | *n* | % Agree | *n* | % Agree | % Change | Sig |
| 1. I feel that I have a number of good qualities. | 1579 | 95.12 | 854 | 95.55 | 0.45 | |
| 2. People my age who drink alcohol are hurting themselves. | 1605 | 85.55 | 864 | 88.43 | 3.37 | * |
| 3. I would encourage a friend who is cutting to get help. | 1599 | 92.56 | 863 | 91.66 | −0.97 | |
| 4. Suicide sometimes seems like a good way to solve my problems. | 1588 | 12.85 | 859 | 14.20 | 10.56 | |
| 5. If I saw someone getting bullied, I would speak up to stop the bullying. | 1585 | 90.09 | 858 | 90.79 | 0.77 | |
| 6. I am hopeful about the future. | 1581 | 93.86 | 861 | 94.43 | 0.60 | |
| 7. It's OK with me if kids my age are smoking pot (marijuana). | 1583 | 17.75 | 860 | 18.26 | 2.84 | |
| 8. True commitment to a boy-/girlfriend means having no other friends. | 1585 | 11.86 | 857 | 11.32 | −4.57 | * |
| 9. Smoking cigarettes and chewing tobacco are very harmful. | 1591 | 79.82 | 856 | 79.79 | −0.04 | |
| 10. I would not be friends with someone who has a mental illness. | 1572 | 17.62 | 855 | 16.73 | −5.08 | |
| 11. Doing well in school is important to me. | 1584 | 96.21 | 860 | 94.42 | −1.86 | |
| 12. Spice (K2), bath salts, and drugs like these are safe to use. | 1571 | 8.59 | 855 | 11.11 | 29.30 | |
| 13. Depression is not real; it's just an excuse for personal failings. | 1564 | 20.40 | 855 | 21.17 | 3.79 | |
| 14. I could talk to an adult if I had a serious problem in my life. | 1579 | 79.35 | 859 | 79.16 | −0.24 | |
| 15. It's possible to be bullied online (like on Facebook or Twitter). | 1582 | 87.55 | 863 | 89.92 | 2.71 | ** |

Statistical significance levels: * $p < 0.05$; ** $p < 0.01$; *** $p < 0.001$. Source: I Got U evaluation surveys, 15 attitudinal outcome measures.

**Table 6.** Pretest vs. follow-up changes in IGU participant attitudes, project year 3.

| Items | Pretest | | Follow-Up | | Pre vs. Follow-Up | |
|---|---|---|---|---|---|---|
| | *n* | % Agree | *n* | % Agree | % Change | Sig |
| 1. I feel that I have a number of good qualities. | 2670 | 95.36 | 1425 | 94.60 | −0.80 | |
| 2. People my age who drink alcohol are hurting themselves. | 2686 | 85.26 | 1424 | 86.17 | 1.07 | |
| 3. I would encourage a friend who is cutting to get help. | 2674 | 92.37 | 1424 | 90.80 | −1.70 | |
| 4. Suicide sometimes seems like a good way to solve my problems. | 2674 | 11.93 | 1422 | 16.03 | 34.40 | *** |

**Table 6.** *Cont.*

| Items | Pretest | | Follow-Up | | Pre vs. Follow-Up | |
|---|---|---|---|---|---|---|
| | *n* | % Agree | *n* | % Agree | % Change | Sig |
| 5. If I saw someone getting bullied, I would speak up to stop the bullying. | 2673 | 91.77 | 1424 | 89.54 | −2.43 | * |
| 6. I am hopeful about the future. | 2672 | 93.56 | 1421 | 91.48 | −2.22 | * |
| 7. It's OK with me if kids my age are smoking pot (marijuana). | 2677 | 16.32 | 1420 | 21.06 | 28.99 | *** |
| 8. True commitment to a boy-/girlfriend means having no other friends. | 2668 | 10.19 | 1419 | 13.25 | 29.95 | ** |
| 9. Smoking cigarettes and chewing tobacco are very harmful. | 2672 | 81.10 | 1417 | 79.96 | −1.41 | |
| 10. I would not be friends with someone who has a mental illness. | 2665 | 18.12 | 1412 | 21.25 | 17.23 | * |
| 11. Doing well in school is important to me. | 2678 | 93.61 | 1418 | 87.66 | −6.36 | *** |
| 12. Spice (K2), bath salts, and drugs like these are safe to use. | 2651 | 9.73 | 1415 | 13.85 | 42.33 | *** |
| 13. Depression is not real; it's just an excuse for personal failings. | 2649 | 21.93 | 1413 | 24.91 | 13.58 | * |
| 14. I could talk to an adult if I had a serious problem in my life. | 2672 | 80.58 | 1421 | 79.17 | −1.75 | |
| 15. It's possible to be bullied online (like on Facebook or Twitter). | 2670 | 89.06 | 1420 | 88.80 | −0.29 | |

Statistical significance levels: * $p < 0.05$; ** $p < 0.01$; *** $p < 0.001$. Source: I Got U evaluation surveys, 15 attitudinal outcome measures.

Comparisons of pretest data gathered during the daylong intervention with follow-up data collected from students four to six months afterward tell much the same story across program years (see Tables 4–6). However, as is often the case in interventions, salutary follow-up results are considerably less robust than those evident at post-test. At follow-up for year 1 (Table 4), students who completed the evaluation instruments continued to report bolstered self-esteem (item 1, $p < 0.001$), and strong risk of harm attitudes toward synthetic drugs (item 12, $p < 0.001$). Other items were either significant in an adverse direction or not significant. Year 2 yielded roughly similar follow-up results. Enduring effects were evident four to six months after the intervention for three of the fifteen evaluation items (Table 5), namely, perceived risk of underage drinking harm (item 2, $p < 0.05$), understandings of healthy romantic relationships (item 8, $p < 0.05$), and online bullying awareness (item 15, $p < 0.01$).

In project year 3, during which dramatic service expansion occurred, follow-up results dropped off precipitously. These results are displayed in Table 6, which reveals nine statistically significant effects. However, all of these effects demonstrate adverse (undesirable) changes in participant attitudes. They all run contrary to intervention aims. Thus, there was not a single salutary follow-up change exhibited by the year 3 participants. In short, as was the case for the year 3 post-test results, the follow-up year 3 results demonstrated the ineffectiveness of an economy of scale approach.

Several alternative explanations beyond the economy of scale approach also merit consideration, although we find these alternatives less than compelling for reasons described here. It is possible, of course, that the differential results observed across years 1–3 were influenced not solely by event size but by evaluation survey attrition (item-specific non-responses or whole survey refusal) from pretest to post-test. Different proportions of students failing to complete post-test surveys across years might bias the results. Increased survey attrition would be expected to produce more commonly observed desirable associations, that is, a positive response bias due to selectivity in the form of a salutary impact creating a motive for participants to complete the post-test survey. Therefore, we calculated survey attrition rates for each item during each project year by dividing the number of post-test responses by the number of pretest responses and then subtracting that quotient (percent) from one to yield a non-response rate for that item. We then calculated mean survey attrition rates for each year by combining all item-specific attrition rates for that year and dividing that figure by the number of

survey items featured in the instrument (15). The mean survey attrition rate for year 1 was 11.77 (minimum = 10.69, maximum = 12.23). The year 2 survey attrition rate of 30.10 (minimum = 29.56, maximum = 30.58) was nearly three times that of year 1. And the year 3 survey attrition rate of 39.07 (minimum = 38.89, maximum = 39.28) was somewhat greater yet.

For two reasons, the foregoing survey attrition rates do not seem to provide a compelling alternative explanation to the economy of scale model we have sought to examine here. First, the key point of distinction in the observation of diminished significant associations was year 3 while the marked increase in survey attrition occurred between years 1 and 2. Therefore, the more pronounced survey attrition changes from years 1 to 2 do not align with the precipitous year 3 diminishment in statistically significant findings. Second, the reasonable expectation of positive response bias with greater survey attrition was not observed. Null findings were most abundant in year 3, the year during which survey attrition was greatest. Although we cannot test this alternative more rigorously, it does seem that survey attrition was not responsible for the changes in results observed across project years. Hence, our application of the economy of scale argument seems to be a compelling explanation for the changes observed across project years. Of course, these are not mutually exclusive explanations. Larger events provide an excellent means for testing an economy of scale model but also invite higher rates of survey incompletion by virtue of their sheer size and the logistical challenges of conducting an evaluation at such venues.

Yet a second alternative explanation for the distinct patterns observed in years 1–2 versus year 3 also requires consideration. Participant demographic characteristics could influence the evaluation results across program years. Quite notably, the inclusion of demographic factors as controls in ancillary analyses did not alter the results. Additional inspection of the data based on returned evaluation surveys revealed that the student groups served across project years were quite similar. Gender and race-ethnicity show some variation across years, while age does not (given the fact that only eighth-graders and tenth-graders were served). During project year 1, the majority of students served were male (52% vs. 48% female students) and white (48% vs. 43% black students). This pattern was largely replicated in project year 3, during which male students (53%) outnumbered their female peers (47%), and the number of white students (44%) eclipsed their African American counterparts (39%). Slight variations were evident in project year 2, during which female students (52%) outnumbered their male counterparts (48%), while black students (47%) eclipsed white students (39%) who were served by the program.

Overall, then, there were generally comparable groups served across program years, and demographic factors did not exert a confounding influence on the patterns reported here. More importantly, differentials across years are most pronounced between project years 1 and 2. As noted above, the types of students served during project years 1 and 3 are somewhat similar. Yet, it is year 3 where we observe null findings, that is, the relative absence of significant associations that had been observed in years 1 and 2. If demographic influences were at work here, we would expect different outcomes to surface in year 2, during which more female and black participants were served. However, we find that results for years 1 and 2 are quite similar, with a dramatic drop-off in impact during year 3. It is worth noting that these two alternative explanations—survey attrition and evaluation sample composition across years—cannot be dismissed altogether, as survey refusals may have occurred in a non-random fashion influenced by demographic backgrounds. We concede that we cannot test this prospect in a rigorous fashion.

A third and final alternative explanation for the disparate findings observed in years 1–2 versus year 3 concerns implementation fidelity (i.e., consistency of delivery). We do not have statistical data to verify implementation fidelity. However, efforts were made to ensure that no appreciable changes in programmatic content or intervention delivery occurred across project years. In fact, a curriculum was developed and pilot-tested prior to the evaluation to ensure consistency in content and delivery. Moreover, the speakers did not change appreciably from one program year to the next, and all

had previous experience delivering the intervention prior to the evaluation. Thus, implementation "drift"—that is, lack of adherence to delivering the program as planned—is an unlikely influence.

## 4. Discussion

This study reported the evaluation results of a three-year youth behavioral health program implemented for eighth and tenth-grade students in central Mississippi. The intervention, I Got U (IGU), coupled risk prevention with mental health promotion through a single-day immersion program that proved attractive to schools because it did not detract significantly from class time. Given the severe social disadvantages confronted by youth in Mississippi, the implementation setting provided a stringent test of the program's effectiveness. Programmatic changes enlisted during year 3 of the intervention also provided an excellent opportunity to explore the efficacy of applying an economy of scale approach. As described here, demand for the program increased dramatically in year 3, but speaker availability remained limited. Therefore, the project team made the decision to move the intervention to larger venues with no commensurate change in the number of events offered. The program expanded dramatically in year 3, when it served more than three times the number of students. Thus, events that had been capped at 175 attendees during program years 1 and 2 changed significantly in size but not in content during year 3, with one event during that last year serving nearly 800 students. Given the pressure to bolster service numbers in current interventions, the lessons learned from IGU are instructive.

During the first two years of the program, the evidence reveals that IGU was highly effective at changing young people's attitudes on a range of factors (e.g., self-esteem, perceived risk of drug use, healthy romantic relationships). As a single-day intervention, pretest/post-test survey outcomes had to be restricted to attitudinal measures. Behaviors could not reasonably change during a same-day pretest/post-test administration.

Table 7 provides a summary of the number of salutary changes observed across survey types and program years. Ten desirable (salutary) changes were evident from pretest to post-test during program year 1, while seven such changes were detected during program year 2. During program years 1 and 2, some changes remained evident at follow-up four to six months after the intervention date. Two such sustained changes were evident during year 1 and three persistent effects were detected during year 2. Thus, as is common with interventions of this sort, sustained changes at follow-up were not as plentiful or robust as those evident as post-test. The general lack of sustained changes might also be related to the fact that the intervention is a day-long assembly-style immersion rather than a multi-event class.

**Table 7.** Number of desirable significant changes in IGU participant attitudes, years 1–3.

| Program Year | Pretest to Post-Test | Pretest to Follow-Up |
|---|---|---|
| **Program year 1:** Small venues; attendee cap applied | 10 | 2 |
| **Program year 2:** Small venues; attendee cap applied | 7 | 3 |
| **Program year 3:** Large venues; economy of scale applied | 1 | 0 |

Source: I Got U evaluation surveys, 15 attitudinal outcome measures.

While strong evidence of program effectiveness was observed during years 1 and 2 of IGU, the economy of scale model applied during year 3 clearly undermined the effectiveness of the program. During this final year in which the program expanded its service numbers without changes in content, speakers, number of events, etc., only one salutary attitudinal change was evident from pretest to post-test. No salutary changes at all were observed at follow-up. Thus, Table 7 clearly reveals diminished effectiveness in year 3. These lackluster results during the final program year indicate the program was effective when it was serving small groups of participants, but lost its effectiveness with its dramatic expansion.

Why might the economy of scale approach have failed in year 3? Despite the scarcity of research on economies of scale in behavioral health service provision, there is some evidence that scaling up an intervention is not a straightforward endeavor. Rather, transitioning from a small-scale intervention to a large-scale program often requires serious reconsideration of the content, structure, and delivery of the intervention [27]. A strictly consistent approach to implementation in scaling up may, therefore, not always be the best strategy [28]. For instance, an intervention that is significantly expanded to achieve economy of scale aims may require the introduction of breakout groups that preserve opportunities for interpersonal interaction and the cultivation of rapport that is not possible with very large groups. Understandably, innovations such as breakout groups were not attempted with IGU in year 3. In an effort to retain fidelity to the project as it was initially conceived and funded, programmatic content, structure, and delivery remained constant even as the participant base was expanded and the venue was enlarged during year 3. This well-intentioned effort had unforeseen consequences with respect to the impact of the project, but also provided an important cautionary lesson about the possible perils of scaling up.

It would also seem that an economy of scale approach should be accompanied by an awareness of difficult trade-offs with respect to programmatic expansion. To be sure, economies of scale can be more efficient in terms of numbers served. However, by their very nature, they may pose threats to programming quality, and some of those threats may only be evident in hindsight. In a large intervention targeted at young people with ready access to mobile technology, the quality of attention that participants are able to give the intervention activities is likely to be undermined. The temptation to send and receive texts, play games on phones, or engage in other distracting behaviors is much greater in large groups that promise a degree of anonymity. Moreover, group management in a larger venue with triple the attendees poses a significant challenge when compared with a smaller, more intimate setting where mutual monitoring structurally imposes an intrinsic form of social control. Finally, venue space matters in significant ways. In the larger IGU auditorium-style venue used during year 3, speaker-student interaction was more limited even though the programmatic content remained the same. In these larger events, speakers were on a stage unable to travel into the space inhabited by attendees. This arrangement is likely to have inhibited the rapport and interpersonal connection that was fostered by smaller events that were capped at 175.

## 5. Conclusions

This study revealed that a youth behavioral health promotion program called I Got U was highly successful in years 1 and 2, thereby providing evidence to underscore its successful delivery to a relatively small number of students, generally less than 200 per intervention. However, the intervention could not be straightforwardly expanded based on an economy of scale programming model despite the service delivery proficiency gains that accrued by year 3. This conclusion does not suggest that an economy of scale approach is doomed to failure in behavioral health programming. Rather, it only indicates that strategic choices, some of which may involve significant revisions in service delivery protocols, could facilitate a program's expansion while preserving its effectiveness.

This lesson is a valuable one in an age where the number of clients served is commonly considered an important benchmark of program success. In some types of programs, particularly those targeted at youth, the number of clients served should not be the litmus test of success. This evaluation of I Got U underscores the benefits associated with delivering an intervention to smaller cohorts of students so that student attention and speaker-student engagement are maximized. Larger groups of youth program attendees create an environment in which norms of self-monitoring and social connection give way to detachment and distraction. The evidence presented here clearly demonstrates the effectiveness of I Got U provided the number of clients served does not cross the critical threshold of about 200 students at a single intervention. Good things do indeed come in small packages.

**Author Contributions:** The research team consisted of J.P.B. (lead evaluator), associate evaluator (X.X.), and assistant evaluator (J.S.A.). J.P.B. wrote the majority of the manuscript with select contributions from X.X., J.S.A., D.F., and F.J.J. J.P.B. led instrument development with input from X.X. X.X. was responsible for the entry and analysis of all data while developing tables for dissemination. All authors have contributed substantially to this manuscript. All have read and approved this manuscript for submission.

**Acknowledgments:** The authors credit the anonymous reviewers of J with many astute insights that improved the quality of this article, although the authors alone accept responsibility for any errors of fact or interpretation. Select program funds were used with approval to support this independent evaluation. The evaluation funds supported instrument development, data collection, data management, data analysis, report generation, and results dissemination. A participatory, utilization-focused approach to evaluation was employed. Consistent with this approach, the lead evaluator developed all instruments. Feedback on instruments was then solicited from the program team prior to field-testing them and ultimately utilizing the field-tested measures. All facets of data entry, analysis, and reporting were conducted separately and independently by the evaluation team.

**Conflicts of Interest:** The authors declare no conflict of interest. No industry or proprietary funds were used to conduct this evaluation. As is common with independent evaluations, a small portion of program funds was used to support evaluation activities. The program team members are named as coauthors in this manuscript given reviews and suggested revisions to instrument development. Such input is consistent with a utilization-focused approach to evaluation that is common in the field.

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
