# Peer review of "Good Things in Small Packages? Evaluating an Economy of Scale Approach to Behavioral Health Promotion in Rural America"

_2571-8800, doi:10.3390/j1010006_

Round 1

Reviewer 1 Report

This paper describes evaluation of a daylong intensive immersion program for youth risk prevention and mental health promotion implemented in differing conditions over a three year period.  Changes in youth self-report appear to decrease from year 1 to year 3.  The authors note that the program was delivered to much larger groups in year 3 with all other aspects held constant, and propose that this “economy of scale” approach is what led to the decreased change in year 3.  The paper is very clearly written with an engaging voice; I quite enjoyed reading it.  I also commend the authors on writing a compelling background section that makes a strong case for the present study.  The main limitations of the study design are a lack of comparison group and inability to link pre-post data, which the authors note, but these limitations reflect typical challenges in program evaluations in which funding is primarily for programmatic activities and so I don’t see this to be a flaw in design, rather just a limitation that is well-noted.  Instead, I have a few other main concerns regarding risk of selection bias/attrition and results interpretations.

First, the primary concern that I have is that the increased attrition rates over time may be an alternative explanation for the change, and that this is not acknowledged in the manuscript.  I’m a bit perplexed by the implications of this; on the one hand, it may be that the increased attrition itself is an indication of the decreased engagement in the larger Year 3 implementation format, in which case the attrition itself almost proves the point.  On the other hand, it seems likely that the larger attrition would be concentrated among those for whom the program had limited impact (due to lower engagement), in which case I would expect to see larger changes in Year 3 as the results would be restricted to those who were most likely to be positively engaged in the program.  But it seems if the higher attrition was to be attributed to the larger implementation in year 3 it should be similar in years 1 and 2, yet it appears that there was also greater attrition in year 2 than in year 1, which suggests that something else might be going on.  (Note: in the methods, only pooled attrition rates are provided; although readers can calculate by item, given the potential significance of this issue attrition rates should be clearly provided by year, or at least mention given to the fact that attrition increased each year. For example, for the first item the attrition was 11.7% in Y1, 29.7% in Y2, and 39% in Y3.) A couple of thoughts are the potential of a cohort effect, and also that the program served 8th and 10th graders over three years – is it probable that many of the 10th graders had already participated in the program in 8th grade and were getting a repeat dose?  It would be helpful to make this clearer, and overall I suggest that the authors more thoroughly address the attrition issue both in the methods/results and also discuss the implications for this in the discussion.

Another serious concern I have is the apparent issues with the p-values including repeat testing, use of one-tailed tests, and the authors’ inclinations to rely too heavily on statistical significance rather than actually meaningful differences.  The methods describe a total of 90 t-test comparisons (pre-post and pre-follow-up for each of 15 items over three years), meaning there is a huge issue with repeat testing that has not been accounted for. This is problematic because in the results (for example, the paragraph beginning in Line 21), the authors rely solely on the statistical significance to interpret the change. For example, they report “bolstered self-esteem” at follow-up for Year 1, when in fact the % change was .33. In Year 3, the % change for the same item was .45 but was not statistically significant, so this similar pre-post change is interpreted differently across the two years.  (Note: I also don’t see how the .33 is so highly significant and suggest the numbers be rechecked throughout).  There is also a statement that “Other items were either significant in an adverse direction or not significant.” Yet this is confusing due to the use of one-tailed rather than two-tailed tests.  Note that given the relatively large number of changes that were not in the anticipated direction, I would caution against the use of one-tailed tests in order to better ascertain the potentially adverse effects of the program. Regardless of test used, there needs to be more acknowledgement and discussion of apparent adverse changes.

A couple more minor comments:

1) it’s not clear in the methods: the tables report the proportional percent change, but I’m assuming the t-tests were conducted on the raw percentages, right?  It would be helpful to clarify this.

2) I’m not clear why the 4-point responses recoded to be dichotomous.  This seems like it has the potential to miss more minor changes that could be potentially meaningful (eg: switching from “agree” to “strongly agree”) and that could be better captured with leaving the 4-point responses as-is and calculating a mean score.

Author Response

We have created a document detailing our response to Reviewer 1, to whom we are grateful for such thoughtful and helpful comments. See the attached file.

Reviewer 2 Report

This is a very interesting paper with is very novel to me. The paper is very well written, and the lit review is good. Not many studies have tested economy of scale paradigm to behavioral promotion programs. The study used a natural experiment data, on three years of an intervention. The first two years, the intervention groups were small, and the last year, it was larger groups. The results show larger effects in the first years, and the author conclude

These are some comments:

1- I would prefer to see more methodological information in the paper. What are the references for these measures? What is eligibility? Drop out?

2- What are the demographic factors across years? Are the groups comparable? Even if you control for them, some groups may be differently ready for a change.

3- What are the statistical tests that are used for inference? A revision is needed for the statistical section of the paper.

4- The measures are Likert, but analyzed as a dichotomous variable. What happens to the results if you reanalyze the data as rank order variables?

5- The individuals are nested , as they have received the intervention in groups. So, authors can not use statistics that do not consider similarities in . One way is to run multi-level analysis.

6- Some of the items go together. So, it does not make a lot of sense to have hundreds of outcomes, because these are not independent outcomes. Run factor analysis and show us how many domains do these items belong to?

7- The assumption is that one size fits all. That means,  all people in year 1 and year 2 and year  3 are similar. However, males and females and young and older youth may differently respond based on year.

8- What if the poor effects over time is not because of the n in each group but because of the voltage drop. Show us fidelity of the program is the same over the years. What if the people (the people who administered or ran or lectured) had different efficacy?

Author Response

We have created a document detailing our response to Reviewer 2, to whom we are grateful for such thoughtful and helpful comments. See the attached file.

Round 2

Reviewer 1 Report

I appreciate the authors' responsiveness and efforts to incorporate the suggestions provided by the reviewers, which I believe have clarified most issues raised.  Regarding the issue of repeat testing, I think the authors misunderstood what I meant, as their response was to describe the generation of the instruments.  That's all fine.  What I meant is that there is an issue with inflated risk of Type 1 error due to multiple comparisons - i.e., the kind of problem that is usually corrected by using a more conservative p-value than a .05.  But the authors do indicate the strength of the p-value by varying number of asterisks so fair enough, the reader can certainly decide whether to interpret the results of those with marginal p-values a bit more cautiously.  Otherwise, nice work.

Reviewer 2 Report

The revision is very satisfactory. I should congratulate the authors for their outstanding paper.